# Extraction, Isolation and Identification of Low Molecular Weight Peptides in Human Milk

**DOI:** 10.3390/foods11131836

**Published:** 2022-06-22

**Authors:** Hailong Xiao, He Jiang, Haiyun Tu, Yanbo Jia, Hongqing Wang, Xin Lü, Ruosi Fang, Gongnian Xiao

**Affiliations:** 1Key Laboratory of Agricultural Products Chemical and Biological Processing Technology, Zhejiang University of Science and Technology, Hangzhou 310023, China; 11413005@zju.edu.cn (H.X.); rsfang@zust.edu.cn (R.F.); 2Hangzhou Institute for Food and Drug Control, Hangzhou 310017, China; jh1657726930@sina.com (H.J.); haiyuntu0928@sina.com (H.T.); 13967170634@139.com (Y.J.); cairouer0524100@sina.com (H.W.); 3Zhejiang Market Bureau Supervision Key Laboratory of Dairy and Dairy Products, Hangzhou 310017, China; 4College of Food Science and Engineering, Northwest A&F University, Xianyang 712100, China; xinlu@nwsuaf.edu.cn

**Keywords:** human milk, free low molecular weight peptides, identification

## Abstract

Human milk contains numerous free low molecular weight peptides (LMWPs), which may play an important role in infant health and growth. The bioactivities of LMWPs are determined by their structures, especially the amino acid sequences. In the present study, 81 human milk samples were collected and purified by cation-exchange solid-phase extraction (SPE). Ultra-high performance liquid chromatography coupled to quadrupole time-of-flight mass spectrometry (UPLC-QTOF-MS) was used for the separation and detection of free LMWPs in human milk. A total of 56 LMWPs were identified and quantified. These LMWPs were mainly derived from 3 regions of β-casein, which were the amino acid fragments of 16–40, 85–110, and 205–226. The predominant LMWPs were RETIESLSSSEESITEYK, RETIESLSSSEESITEYKQKVEKVK, ETIESLSSSEESITEYK, TQPLAPVHNPIS, and QPLAPVHNPISV with molecular weights of 2247.9573, 2860.2437, 2091.8591, 1372.7666, and 1271.7212, respectively. The results indicated that the technique based on SPE and UPLC-QTOF-MS might greatly facilitate the analysis of LMWPs in human milk.

## 1. Introduction

The importance of proteins in the diet has been increasingly acknowledged as a result of new scientific findings in the field of nutrition over the last two decades. The value of proteins as an essential source of amino acids is well documented, but recently it has been recognized that dietary proteins exert many other functionalities in vivo by means of biologically active peptides [1], which have been defined as specific protein fragments that have positive impacts on body functions or conditions and may ultimately influence health [2]. The size of bio-active peptides may vary from 2 to 30 amino acid residues; these peptides are also called low molecular weight peptides (LMWPs). Despite being studied for years, the LMWPs have never attracted enough attention. In fact, this peptide fraction is a key factor that regulates many genes and participates in various pathways of disease [3,4]. Recent research has found a large number of functional endogenous peptides in breast milk, thus making human milk a carrier of biochemical messages [5,6,7,8,9]. Over 300 naturally produced peptides were identified originating from the protein composition of breast milk, and the antimicrobial activity of the peptide mixture was firstly considered [5].

Milk components, including lipids, proteins, and other nutrients, are affected by various factors such as species, season, environment, and lactation periods [10,11,12]. Over the past decades, a number of LMWPs with specific bioactivities have been identified in bovine milk, including calcium-binding phosphopeptides (CCPs) [13], angiotensin-converting enzyme (ACE), inhibitory [14,15], antibacterial [16,17], antioxidative [18,19], immunomodulatory [15,20], opioid-like [21], and cell growth-stimulating effects [22]. These peptides are also believed to exist in human milk and play similar roles [23], and such bioactive functions are indispensable for infants whose cardiovascular, digestive, endocrine, immune, and nervous systems are still not fully developed.

The bioactive functions of LMWPs are based on their inherent amino acid composition and sequences. The study of detection and identification is the first step toward understanding what specific functions peptides can exert in human milk. So far, the number and location of peptides derived from different proteins in human milk have been reported based on Chip-QTOF (Chip-quadrupole time-of-flight mass spectrometry) analysis [24,25]. However, the common LMWPs and their variation in human milk among different individuals remain to be revealed. The aim of the present study was to determine the common LMWPs among 81 human milk samples, which were collected from eastern China. For this purpose, we used a novel, streamlined, high-throughput analytical approach optimized to explicitly capture and identify the complete set of peptides produced by human in vivo proteolytic digestion of breast milk.

## 2. Materials and Methods

### 2.1. Reagents and Equipment

Ammonia (AP) and hexane (AP) were bought from Huadong Medicine Co., Ltd. (Hangzhou, China), acetonitrile (HPLC-grade) and methanol (HPLC-grade) were acquired from Fisher Scientific Ltd. (Rathburn, Walkerburn, UK), formic acid (solution 96%, HPLC-grade) was acquired from Tedia company, Inc. (Fairfield, OH, USA). Oasis^®^ MCX extraction cartridges (60 mg/^3^ mL) and the manifold system were purchased from Waters (Milford, MA, USA). Leu-enkephalin (LE) was obtained from Sigma-Aldrich Co., LLC. (St. Louis, MO, USA) Samples of human milk were obtained from local hospitals. Hitachi L-8900 amino acid analyzer was purchased from the Hitachi company (Tokyo, Japan).

### 2.2. Sample Collection and Preparation

A total of 81 human milk samples (50 mL) were collected in eastern China; these samples were from 15 to 210 days of lactation. All donors were healthy and gave birth to healthy infants and were aged from 18 to 36 years old. All samples were mechanically milked during the middle lactation stage into sterile polystyrene containers, frozen immediately, and stored at −80 °C until use to prevent undesired proteolysis. A 5 mL homogenized human milk sample was mixed with 5 mL formic acid solution (1% in purified water, *w*/*v*) and 5 mL hexane. The mixture was shaken gently for 1 min to homogenize. After centrifuging at 3000 rpm for 10 min, the fat (hexane layer) was removed, and the supernatant was filtered through an ultrafiltration membrane to remove bacteria and cell debris. 

### 2.3. Solid Phase Extraction

A 10 mL sample solution was passed through the cartridges at 1 mL/min. The cartridges were cleaned with 5 mL 1% formic acid and dried under vacuum. They were eluted with 2 mL methanol containing 5% (*v*/*v*) NH_3_⋅H_2_O. The elutes were collected, evaporated to net dry, and dissolved in 1 mL mobile phase (A), and 5 μL was injected into the UPLC-QTOF system for analysis. 

### 2.4. Chip-QTOF Analysis

UPLC was performed in a Waters Acquity Ultra-high Performance Liquid Chromatography (UPLC) system (Waters, Milford, MA, USA) equipped with an autosampler and binary solvent delivery system. The chromatography was performed on a Waters Acquity BEH C18 column (2.1 mm × 100 mm, 1.7 μm, Waters, Milford, MA, USA). The mobile phase consisted of (A) 0.1% formic acid in water and (B) ACN containing 0.1% formic acid. Reversed-phase separation parameters were initially 5% B (3 min) followed by a linear gradient from 5 to 40% B over 35 min and subsequently a ramp from 40 to 90% B over 5 min. After a hold at 90% B for 5 min and a ramping step down to 5% B over 1 min, the system was re-equilibrated at 5% B for 10 min. Leucine-enkephalin (0.1 μg/mL) was added as an internal standard. Mass spectrometry was performed using a Waters QTOF Premier (Micromass MS Technologies, Manchester, UK) equipped with an electrospray ionization (ESI) source operating in positive ion mode. MS analysis was performed using positive electrospray ionization mode. The nebulization gas was set to 600 L/h at a temperature of 350 °C, the cone was 50 L/h, and the source temperature was 100 °C. The capillary voltage and cone voltage were set to 3000 V and 30 V, respectively. The QTOF acquisition rate was set to 0.2 s, with a 0.01 s inter-scan delay. Argon was employed as the collision gas at a pressure of 7.066 × 10^−3^ Pa.

The MS^E^ experiments were operated with the premier quadrupole in a wide pass mode with the collision cell operating at 6 V and a ramp high energy from 25 V to 65 V, which was set to produce multiple fragment ions for QTOF analysis. The MS/MS experiments were carried out by setting the premier quadrupole to allow ions of interest to pass, followed by ramping collision energy from 25 V to 65 V to produce abundant product ions. Before sample testing, the QTOF mass spectrometer was tuned and calibrated following the manufacturer’s instructions. Leucine-enkephalin (*m*/*z* 556.2771 in positive ion mode) was applied for real-time calibration in the LockSpray mode at a concentration of 200 ng/mL and an infusion flow rate of 10 μL/min.

### 2.5. Data Analysis

The UPLC-QTOF-MS data of all determined samples were analyzed by MarkerLynx software (Waters, Milford, MA, USA). For data collection, the parameters were set as follows: retention time ranging from 2 to 35 min, mass ranging from 100 to 2000 Da, and mass tolerance at 0.02 Da. Sequencing analysis of low molecular weight peptides was performed by ProteinLynx Global Server 2.5 software (PLGS, Waters, Milford, MA, USA). The low energy threshold, elevated energy threshold, and intensity thresholds were set to 250.0 counts, 100.0 counts, and 750.0 counts, respectively. The data bank was human milk protein which was downloaded from the National Center for Biotechnology Information (NCBI) database. The peptides confirmations were performed by QTOF MS/MS, combined with peptides sequencing using BioLynx software (Waters, Milford, MA, USA).

### 2.6. Ethical Statement

The study protocol was approved by the Ethics Committee of the College of Biosystem Engineering and Food Science, Zhejiang University (No. 2012006). Written informed consent was obtained from all study participants. The study conforms to the principles outlined in the Declaration of Helsinki.

## 3. Results

### 3.1. Extraction and Separation of LMWPs

Hexane was used to remove fats and precipitate proteins, and an ultrafiltration membrane allowed the peptides less than 5000 Da to remain. Cation-exchange solid-phase extraction was used to purify LMWPs based on their positive charged nature in acid solution. The purified samples were separated with UPLC, and the chromagram showed that the extraction contained abundant peaks (Figure 1).

### 3.2. Sequencing Analysis and Quantification of LMWPs

Samples were analyzed by QTOF after UPLC separation. Most of the peptides had multiple charges (Figure 2A, peak of 572.3053 and 686.8523), while a few LMWPs had single charges (Figure 2A, peak of 1143.6125 and 1372.7349). Peaks data were collected by the MSE model, which enabled the simultaneous acquisition of both LC/MS and fragmentation data from a single experiment. LMWPs were sequenced by PLGS based on accurate molecular weight and human milk protein database (Figure 2B). Sequence confirmation was performed by QTOF MS/MS with peptides sequencing of BioLynx software.

A total of 56 common LMWPs were identified and quantified in 81 samples (Table 1). Most peptides were fragments of β-casein precursor, while a few were from the k-casein precursor. The protein precursor, residue position, sequence, mass (MH+), charge state, retention time (RT), and concentration are listed in Table 1.

### 3.3. Variations of Main LMWPs 

A total number of 16 LMWPs whose concentration was more than 0.1 μg/mL were picked out for the analysis of variation. Of these, 75 samples were evenly divided into 3 groups according to different lactation times (10–60, 70–130, and 180–230 d) for the analysis of the relationship between LMWPs concentration and lactation time, and 69 samples were divided into 3 groups by the mother’s age (19–22, 25–28, and 32–36 years old) for the variation of LMWPs. The mean LMWPs concentration of each group was calculated. The results showed that the variation of LMWPs concentration at different lactation times is not significant (Figure 3A). There was an obvious change in LMWPs with childbearing age, and the concentration of the younger group (19–22) was less than the other two groups (*p* < 0.05) (Figure 3B).

## 4. Discussion

Human milk is complex and contains fats, proteins, carbohydrates, and other essential nutrients that could interfere in the detection of LMWPs, so a proper extraction method is necessary. After removing fats and proteins, we applied a cation-exchange solid-phase cartridge to clean the extract, which could exclude most uncharged organic compounds, and the results showed that the SPE was useful and simple for extraction of LMWPs in human milk. 

The accuracy of mass was the key to the successful identification of peptides. However, there would be a slight fluctuation in the signal of mass spectrometry due to the impact of the laboratory environment, which would lead to erroneous results in the following peptide sequencing. One solution was real-time calibration with a known mass standard. In this study, Leucine-enkephalin for real-time calibration was applied to ensure the mass accuracy. On the other hand, mass accuracy and resolution of the QTOF-MS analyzer were not enough to achieve an unequivocal identification since the same mass of peptide fragments would have different amino acid sequences. This could be solved by comparing with the sequence information of existing proteins. The human milk protein database downloaded into PLGS from the NCBI database was used as peptides templates to confirm the identities of some proposed peptide sequences. In general, the determination of natural LMWPs by real-time calibration of the QTOF-MS method combined with the protein database has been suggested to be an ideal method, which was especially suitable for large-scale identification of small peptides when standards were not available.

The quantitation of proteins and peptides in complex biological systems is one of the most challenging areas of proteomics. Due to the lack of standards or biomarkers for each component, accurate quantitation of LMWPs is difficult. We demonstrated a simple method by comparing the intensity of LMWPs with that of the Leucine-enkephalin standard. Despite the possible deviation, the approximate concentration and variation of LMWPs could be calculated conveniently.

Among all the detected small peptides, casein phosphopeptide (CPP) had the highest concentration, which reached about 10–15 μg/mL in breast milk, with “RETIESLSSSEESITEYK”, “ETIESLSSSEESITEYK”, “RETIESLSSSEESITEYKQKVEKVK” as the major form [26]. The main function of CPP was promoting calcium absorption, thus making it essential for the rapid growth of infants. Antibacterial peptides were another abundant peptide in breast milk, mainly in the form of “TQPLAPVHNPISV”, “QPLAPVHNPISV”, “PLAPVHNPISV”, “NPTHQIYPVTQPLAPVHNPISV”, etc. [27], the content was about 5–10μg/mL. They were important for resisting bacterial invasion due to the imperfect infant’s immune system. The abundance of CPP and antimicrobial peptides was closely related to the growth, development, and health of infants and young children. Therefore, these data could provide a certain reference for the improvement and optimization of infant formula milk powder. 

In addition, we also detected angiotensin-converting enzyme inhibitory peptides (HLPLP, ENLHLPLPL, LPLPLL, etc.) [28,29], antioxidant peptides (NPYVPRT, VPYPQR, SVPQPK, etc.), proline Endopeptidase inhibitory function (PEPi), (YPFVEPIPYG, PFVEPIPY, VEPIPYG, etc.) [28,30], Opioid Agonist (YVPFP, YVPFPPF, YPFV, etc.) [31,32,33,34], Immunostimulating peptides (VEPIPY, GFL, GLF, etc.) [34,35], and other related small peptides, the effects of these peptides on the growth and health of infants and their mechanism of action remain to be further studied.

In general, the results from the present study indicated that the technique based on SPE and UPLC-QTOF-MS is a simple way to extract, isolate and identify LMWPs in human milk and may greatly facilitate ongoing efforts to understand their composition and structure.

## Figures and Tables

**Figure 1 foods-11-01836-f001:**
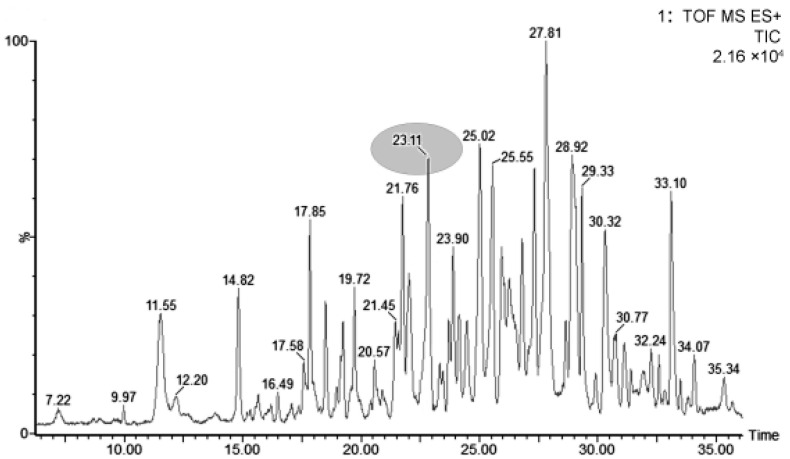
Chromagram of low molecular weight peptides extracted from human milk by SPE.

**Figure 2 foods-11-01836-f002:**
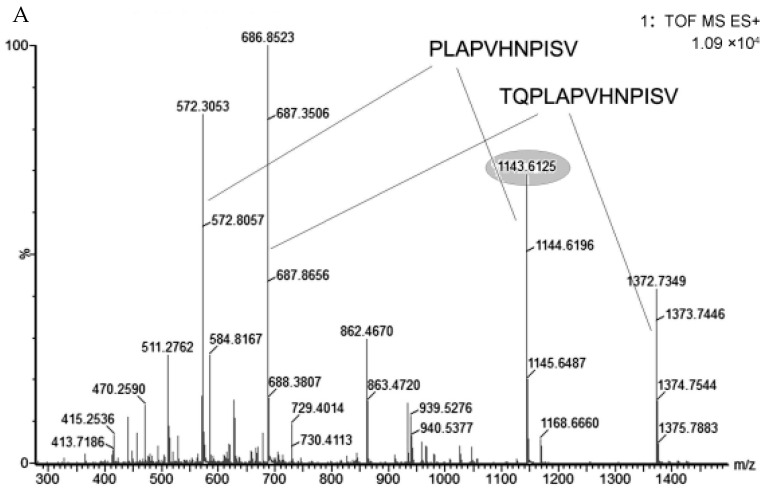
(**A**) Mass spectrum of LMWPs in human milk (RT: 23.11). Peak of 572.3053 and 686.8523 were peptides with single charges, and peak of 1143.6125 and 1372.7349 have 2 charges according to the *m*/*z* interval between isotopes. (**B**) Peptide (Peak 1143.6152 in Figure 2) was sequenced by PLGS based on MS^E^ experiment. The peptide sequence was PLAPVHNPIS, which was a fragment of beta casein precursor in human milk.

**Figure 3 foods-11-01836-f003:**
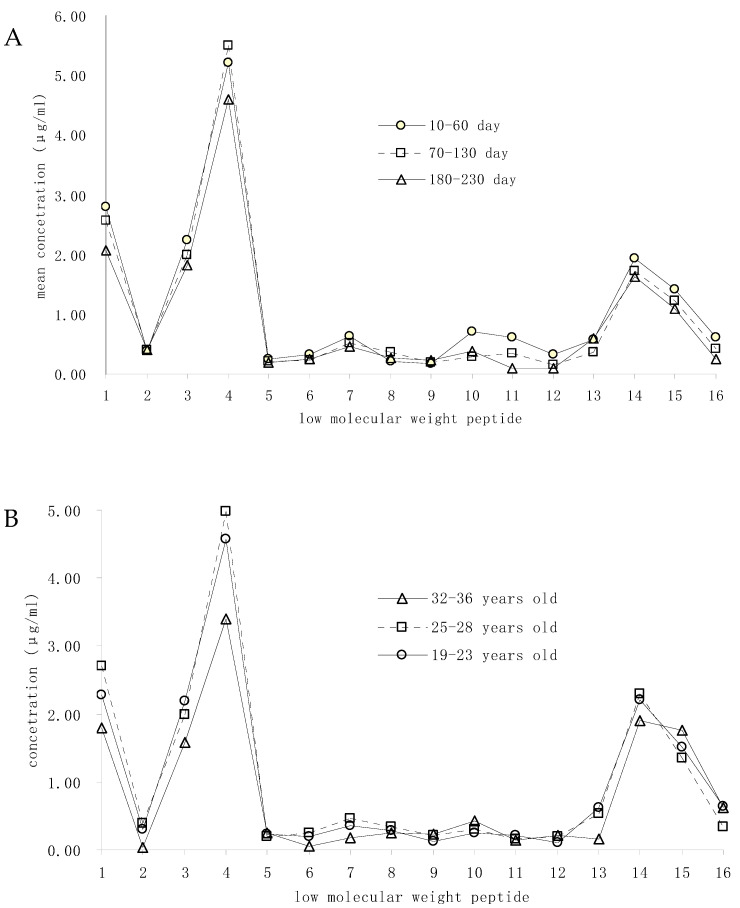
(**A**) The variation of main LMWPs with different lactation times. (**B**) The variation of main LMWPs with different childbearing ages. 1: RETIESLSSSEESITEYK; 2: RETIESLSSSEESITEYKQKVEK; 3: RETIESLSSSEESITEYKQKVEKVK; 4: ETIESLSSSEESITEYK; 5: ETIESLSSSEESITEYKQKVEK; 6: SITEYK; 7: TEYK; 8: QPAVVLPVPQPEIMEVPKAKDTVYTKG; 9: VVLPVPQPEIMEVPK; 10: VVLPVPQPEIMEVPKAKDTVYT; 11: LPVPQPEIMEVPK; 12: PQTLA; 13: NPTHQIYPVTQPLAPVHNPISV; 14: TQPLAPVHNPISV; 15: QPLAPVHNPISV; 16: PLAPVHNPISV.

**Table 1 foods-11-01836-t001:** The common LMWPs identified in human milk.

Protein Precursor	Residue	Sequence	MH+ (Da)	Charge State	RT (min)	Concentration μg/mL
*β-casein	16–33	RETIESLSSSEESITEYK	2247.9573	2	19.16	5.315 ± 1.267
β-casein	16–38	RETIESLSSSEESITEYKQKVEK	2700.3093	3	18.97	0.323 ± 0.109
β-casein	16–40	RETIESLSSSEESITEYKQKVEKVK	2860.2437	4	17.77	2.044 ± 0.813
β-casein	17–33	ETIESLSSSEESITEYK	2091.8591	2	19.71	4.065 ± 1.215
β-casein	17–38	ETIESLSSSEESITEYKQKVEK	2544.0308	2	27.84	0.170 ± 0.083
β-casein	23–33	SSSEESITEYK	1259.585	2	22.86	0.003 ± 0.001
β-casein	27–33	ESITEYK	869.4235	1	22.43	0.093 ± 0.037
β-casein	28–33	SITEYK	740.378	1	17.91	0.210 ± 0.069
β-casein	30–32	TEY	412.2193	1	21.09	0.019 ± 0.005
β-casein	30–33	TEYK	540.2654	1	22.43	0.489 ± 0.071
β-casein	31–33	EYK	439.218	1	19.2	0.042 ± 0.009
β-casein	85–103	AQPAVVLPVPQPEIMEVPK	2042.168	2	38.86	0.028 ± 0.006
β-casein	86–103	QPAVVLPVPQPEIMEVPK	1971.1284	2	38.82	0.048 ± 0.015
β-casein	86–112	QPAVVLPVPQPEIMEVPKAKDTVYTKG	2934.5535	3	37.29	0.274 ± 0.067
β-casein	89–103	VVLPVPQPEIMEVPK	1674.97	3	38.08	0.177 ± 0.033
β-casein	89–110	VVLPVPQPEIMEVPKAKDTVYT	2453.3762	3	37.79	0.34 ± 0.086
β-casein	91–103	LPVPQPEIMEVPK	1476.8384	2	30.94	0.188 ± 0.054
β-casein	94–98	PQPEI	583.3124	1	36.25	0.003 ± 0.001
β-casein	94–103	PQPEIMEVPK	1167.6152	2	38.82	0.028 ± 0.009
β-casein	126–130	FDPQI	601.3356	1	36.91	0.074 ± 0.019
β-casein	127–140	DPQIPKLTDLENLH	1632.8809	2	33.68	0.004 ± 0.001
β-casein	150–154	QQVPQ	564.2857	1	19.99	0.055 ± 0.011
β-casein	154–158	PQPIP	551.3185	1	1.77	0.088 ± 0.023
β-casein	157–161	PQTLA	529.2968	1	19.98	0.164 ± 0.036
β-casein	171–189	SVPQPKVLPIPQQVVPYPQR	2270.33	2	33.42	0.005 ± 0.001
β-casein	176–194	VLPIPQQVVPYPQRAVPVQ	2128.2644	2	36.52	0.030 ± 0.008
β-casein	183–194	VVPYPQRAVPVQ	1352.7789	2	11.64	0.035 ± 0.014
β-casein	190–194	VPVQA	495.2942	1	25.06	0.018 ± 0.005
β-casein	191–196	VPVQAL	608.3761	1	25.06	0.003 ± 0.001
β-casein	196–200	LLLNQ	582.358	1	26.81	0.005 ± 0.001
β-casein	199–203	NQELL	598.3207	1	19.12	0.004 ± 0.001
β-casein	204–207	NPTH	468.2182	1	12.71	0.005 ± 0.001
β-casein	205–208	PTHQ	482.2364	1	28.86	0.002 ± 0.001
β-casein	205–226	NPTHQIYPVTQPLAPVHNPISV	2422.3235	3	33.88	0.319 ± 0.081
β-casein	209–226	QIYPVTQPLAPVHNPISV	1973.1118	2	36.92	0.017 ± 0.005
β-casein	211–226	YPVTQPLAPVHNPISV	1731.9642	2	31.9	0.030 ± 0.007
β-casein	212–226	PVTQPLAPVHNPISV	1568.8969	2	27.53	0.042 ± 0.015
β-casein	213–226	VTQPLAPVHNPISV	1471.8435	2	25.41	0.071 ± 0.023
β-casein	214–218	TQPLA	493.2703	1	7.26	0.006 ± 0.002
β-casein	214–226	TQPLAPVHNPISV	1372.7666	2	23.09	2.59 ± 0.951
β-casein	215–226	QPLAPVHNPISV	1271.7212	2	21.96	1.65 ± 0.432
β-casein	216–226	PLAPVHNPISV	1143.6647	1	36.91	0.460 ± 0.105
β-casein	218–224	APVHNPI	729.4062	1	9.75	0.033 ± 0.006
β-casein	218–226	APVHNPISV	933.5039	2	9.78	0.066 ± 0.015
β-casein	222–226	NPISV	529.2968	1	19.98	0.004 ± 0.001
β-casein	223–226	PISV	415.2347	1	28.6	0.024 ± 0.009
*κ-casein	73–79	NPYVPRT	846.4499	1	18.35	0.046 ± 0.018
κ-casein	82–89	ANPAVVRP	823.484	1	19.01	0.008 ± 0.003
κ-casein	82–91	ANPAVVRPHA	1031.5775	2	21.25	0.037 ± 0.019
κ-casein	83–89	NPAVRP	752.4417	1	17.81	0.026 ± 0.011
κ-casein	83–91	NPAVRPHA	960.5374	2	18.66	0.029 ± 0.015
κ-casein	84–89	PAVVRP	638.403	1	25.34	0.008 ± 0.005
κ-casein	99–106	LPNSHPPT	862.4427	2	26.83	0.018 ± 0.010
κ-casein	99–109	LPNSHPPTVVR	1216.6873	2	22.07	0.089 ± 0.026
κ-casein	100–111	PNSHPPTVVRRP	1356.7617	3	30.15	0.036 ± 0.021
κ-casein	104–116	PPTVVRRNLHPS	1469.847	3	33.02	0.041 ± 0.0013

*β-casein: β-casein precursor, *κ-casein: κ-casein precursor.

## Data Availability

Data is contained within the article.

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
