# Peer review of "Extraction, Isolation and Identification of Low Molecular Weight Peptides in Human Milk"

_foods, 2022, doi:10.3390/foods11131836_

Round 1

Reviewer 1 Report

The paper is interesting by identifying milk peptides from human milk and correlating its profile with lactation time and woman age. However, some important points are necessary before it can be considered:

-   An English revision is necessary to remove typo mistakes, such as “deternminded” in abstract, repeated sentences and make the manuscript more fluid

- Authors must improve abstract and introduction to set clear the novelty and contribution of the work to the field

-        Please provide an abbreviation list or ensure that all abbreviations are defined in the first mention in the text

Please provide reference for item “Chip-QTOF analysis”

-        Fig 1: what the grey circle means?

-        Fig 2b: Authors say “There was an obvious change of LMWPs with 162 childbearing age and the concentration of younger group (18-22)”, however in the figure it is written 19-22 and the lowest peptides content is observed for the oldest group. This needs to be corrected. In addition, are the results statistically different?

-        Please keep figure and figure legends in the same page

-        Results need to be more discussed and compared with literature, authors only used 17 references.

Reviewer 2 Report

The manuscript is written well and meets the journal criteria. However there is room for improvement. 

1) The author should provide the y and b ions or the M+1, M+2 or M+3 ions for each of the peptides. Even Though they have attached the mass spectra, it is not clear which ions represent what?

2) It would be great if the author can include the transitions ions for each or the peptides? 

3) Did the author optimize the extraction procedure for the peptide extraction. They have only used the solid phase cation exchange but did the authors try protein precipitation, liquid liquid extraction or any other extraction technique which can improve the extraction efficacy of each of the peptides. 
